# Coherence Evaluation of Visual Concepts with Objects and Language

**Tobias Leemann**[1], **Yao Rong**[1], **Stefan Kraft**[2], **Enkelejda Kasneci**[1], **Gjergji Kasneci**[1]
[1]University of Tübingen, Tübingen, Germany    [2] STZ Softwaretechnik, Esslingen, Germany
`{tobias.leemann,yao.rong}@uni-tuebingen.de,`
`stefan.kraft@stz-softwaretechnik.de`

## Abstract

Meaningful concepts are the fundamental elements of human reasoning. In explainable AI, they are used to provide concept-based explanations of machine learning models. The concepts are often extracted from large-scale image data sets in an unsupervised manner and are therefore not guaranteed to be meaningful to users. In this work, we investigate to which extent we can automatically assess the meaningfulness of such visual concepts using objects and language as forms of supervision. On the way towards discovering more interpretable concepts, we propose the "**S**emantic-level, **O**bject and **La**nguage-Guided **C**oherence **E**valuation" framework for visual concepts (SOLaCE). SOLaCE assigns semantic meanings in the form of words to automatically discovered visual concepts and evaluates their degree of meaningfulness on this higher level without human effort. We consider the question of whether objects are sufficient as possible meanings, or whether a broader vocabulary including more abstract meanings needs to be considered. By means of a user study, we confirm that our simulated evaluations highly agree with the human perception of coherence. We publicly release our data set containing 2600 human ratings of visual concepts.

## 1 Introduction

A multitude of artificial intelligence (AI) systems are built on top of and heavily rely on visual concepts. This term refers to specific parts of an image that can be attributed a higher meaning and are human-interpretable, e.g., "feet", "stripes" or "water" (Bau et al., 2017). In particular, concept-based explanations (Kim et al., 2018; Koh et al., 2020) have recently gained popularity in computer vision. Akula et al. (2020) showed that giving explanations composed of visual concepts increases user trust over pixel-level explanations (saliency methods). However, it remains an open question how such interpretable concepts can be discovered in the data set. Current, unsupervised approaches do not always yield sensible results (see Fig. 1) and there are no established methods to assess their quality without human effort. Recent theoretical insights show that unsupervised approaches fail to find disentangled representations without additional supervision (Locatello et al., 2019). Objects, which are fundamental for human understanding of visual scenes (Löwe et al., 2020) can be such a form of weak supervision and allow for

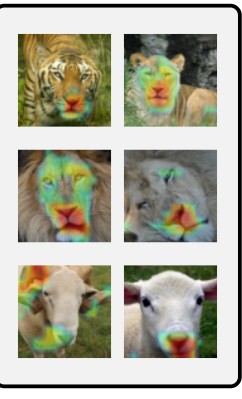
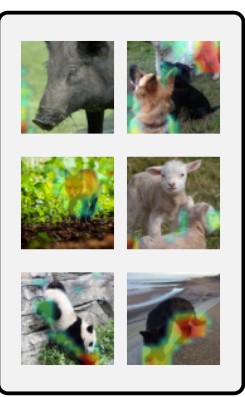

(a) meaningful concept          (b) non-meaningful concept

Figure 1: Two discovered visual concepts by the ConceptSHAP approach (Yeh et al., 2019) on the Animals with Attributes (AwA) data set (Xian et al., 2018). The concept activation map (overlay) indicates the location of the salient parts. While the left one can be easily assigned a meaning ("animal snout") the right one is hard to interpret. In this work, we investigate to which degree we can automatically quantify their meaningfulness.

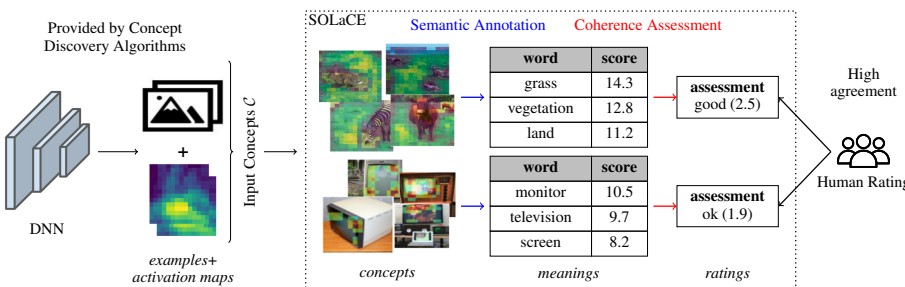

Figure 2: **Overview.** For concepts (consisting of example images and activation maps) that are learned by deep neural networks (DNNs) and automatically discovered post-hoc, **SOLaCE** first provides Semantic Annotations for their meanings in the form of words. Subsequently, it computes a Coherence Assessment for the concepts relying on these annotations and simulates human rating.

learning meaningful representations (Wu et al., 2021). In this work, we follow the similar idea that *meaningful visual concepts have concise descriptions in natural language*, for instance, the name of an object they encode. If such a meaning can be found, it may serve as a certificate for the interpretability of the corresponding concept and allows to separate meaningful from non-meaningful ones. We investigate whether a state-of-the-art object-detection and a joint vision-language model can provide the meanings and allow for an automated coherence assessment on a higher, semantic level. We collect 2600 human assessments[1] of visual concepts and show that (1) a high correlation of $\rho = 0.86$ between the human assessment and our method can be achieved and that (2) using only objects as possible meanings results in higher correlation compared to including more abstract ones.

## 2 RELATED WORK

Many works on conceptual explanations leverage *supervised concepts*. In this case, the set of possible concepts is given by concept annotations in the data set indicating concepts present or absent (Koh et al., 2020; Kazhdan et al., 2020). Unfortunately, most data sets do not have these kinds of annotations, so the important concepts are usually unknown in advance. Therefore, a second strain focuses on *unsupervised concepts*. They are discovered in the data in an unsupervised or weakly-supervised manner and provided as a set of salient examples (Ghorbani et al., 2019; Yeh et al., 2019; Akula et al., 2020). Yet, the problem of how to evaluate the interpretability of the concepts discovered has not been resolved. User studies are the most common way of evaluation but can be prohibitively expensive and prone to design errors (Doshi-Velez & Kim, 2017). Akula et al. (2020) perform an extensive user study to measure *Justified Trust* and *Explanation Satisfaction*. Ghorbani et al. (2019), Yeh et al. (2019) and Laina et al. (2020) also use a human ground truth. Unlike these works, our assessors require no human ground truth, but nevertheless align well with human evaluation. Our framework is most closely related to the *class describability* measure by Laina et al. (2020). The authors propose a modified image-captioning model to automatically find concept descriptions, but do not perform an automated assessment based on these.

Our approach is model-agnostic and efficient as it leverages pretrained object detection and vision-language models with broad coverage. Unlike previous works, it is fully automated. We therefore present a powerful tool to (1) automatically assign semantic meaning in the form of words to arbitrary provided concepts and (2) quantify the intelligibility and coherence of these visual concepts. In turn, this will allow to filter out non-meaningful concepts and to provide better explanations.

## 3 SEMANTIC ANNOTATION AND COHERENCE ASSESSMENT

Building on the related work, we propose the SOLaCE framework to automatically assess the coherence of visual concepts at the semantic level. Our method relies on two modules as depicted in Fig. 2 that can be independently implemented: (1) A *Semantic Annotation Module* to map the concepts to

---

[1]Data set and code are available at https://github.com/tleemann/solace-concepts.

their meanings in the form of a set of words (Sec. 3.2); (2) a *Coherence Assessment Module* which subsequently assesses the coherence of the concepts in a numerical rating (Sec. 3.3). Like previous work (Ghorbani et al., 2019), we consider only single words as meanings in this context. Additional details regarding the implementation of the modules can be found in Appendix B.

## 3.1 VISUAL CONCEPTS

In alignment with prior work, let $\{\boldsymbol{x}_1, \boldsymbol{x}_2, \ldots, \boldsymbol{x}_n\}, \boldsymbol{x}_i \in \mathbb{R}^{H \times W \times C}$ be a set of $n$ example images salient for a certain concept (Kim et al., 2018). Each example has a width of $W$, height of $H$ and $C$ channels. Concepts are *local*, i.e., they are only present in a specific part of an image (cf. also Fig. 1). Mu & Andreas (2020) therefore consider a concept to be a function on the pixels in the input space that evaluates to true, if the concept is present in that pixel. We adopt their approach but allow continuous values in the range $[0, 1]$. Thus, for every example, we also require a concept activation map $\{\boldsymbol{s}_1, \boldsymbol{s}_2, \ldots, \boldsymbol{s}_n\}, \boldsymbol{s}_i \in [0, 1]^{H \times W}$, with values of $1$ indicating the highest correspondence to the concept. Combining both parts, we define a concept $\mathcal{C}$ as a set of $n$ tuples of images and their corresponding activation maps, $\mathcal{C} = \{(\boldsymbol{x}_1, \boldsymbol{s}_1), (\boldsymbol{x}_2, \boldsymbol{s}_2), \ldots, (\boldsymbol{x}_n, \boldsymbol{s}_n)\} \in \mathfrak{C}_n$. In our notation $\mathfrak{C}_n$ denotes the space of concepts given by $n$ examples.

## 3.2 SEMANTIC ANNOTATION MODULE

The annotation module in our framework assigns $k$ words from a user-defined dictionary $\Omega$. Each word comes with a corresponding numerical matching score and represents the semantic *meaning* $\mathcal{M}$ of a visual concept. Thus, $\mathcal{M} = \{(w_j, \gamma_{w_j}) \mid j = 1 \ldots k, w_j \in \Omega, \gamma_{w_j} \in \mathbb{R}\}$, where $w$ denotes the words and $\gamma_w$ the scores. Let $\mathfrak{M}_k$ be the space of meanings with $k$ items, such that $\mathcal{M} \in \mathfrak{M}_k$. Formally, the module is a function $\mathrm{Sem}(\cdot)$ mapping a concept of $n$ examples to its semantic meaning represented by $k$ words: $\mathrm{Sem} : \mathfrak{C}_n \to \mathfrak{M}_k$. We employ two different implementations of $\mathrm{Sem}$:

**Object-Centric Annotation Module (OC).** First, we consider a plain object detection model. We choose FasterRCNN (Ren et al., 2015) for this purpose and use the implementation provided by the Detectron2 framework (Wu et al., 2019). To keep the model as general as possible, we train on the Visual Genome data set (Krishna et al., 2017) with 3434 object categories remaining after classes with insufficient coverage are removed. We use the class names as our dictionary $\Omega$. To assign scores $\gamma_w$, we compute the Soft Intersection over Union (IoU) score (inspired by Bau et al. (2017)) between the object bounding boxes for object type $w \in \Omega$ and the concept activation.

**Joint Vision-Language Annotation Module (JVL).** Because the object detection model has a specific scope due to the data it is trained on (it can only predict classes from the labelled training data set), we investigate a joint vision-language embedding with broader support. Such a model $J : \mathbb{R}^{H \times W \times C} \times \Omega \to \mathbb{R}$ maps both words and images to the same latent space and returns a scalar similarity score. Arbitrary words can be used as inputs. We use the CLIP model (Radford et al., 2021) that has proven highly successful on various language-supported zero-shot learning tasks.

The dictionaries deployed in this case are user-defined and can be large (we use $|\Omega| \approx 10,000$ common words), which is an advantage over the OC model. For each word $w \in \Omega$ and $(\boldsymbol{x}_i, \boldsymbol{s}_i) \in \mathcal{C}$, the embedding model efficiently matches $w$ to example $\boldsymbol{x}_i$ and estimates a score, $y_{i,w} = J(\boldsymbol{x}_i, w)$. We further adjust the scores by using attribution maps of the vision-language model for the words (GradCAM, Selvaraju et al. (2017)) to particularly find relevant words that are aligned with concept activation maps. The alignment over all examples used to compute the final scores $\gamma_w$.

All Annotation Modules return the highest-scoring $k$ words along with their respective fit scores.

## 3.3 COHERENCE ASSESSMENT MODULE

Having identified the meaning $\mathcal{M}$, the coherence assessment module $f : \mathfrak{C}_n \times \mathfrak{M}_k \to \mathbb{R}$ maps the concept and the meaning to a numerical interpretability rating. Note that some implementations of assessment functions only rely on either the meaning or the concept examples and ignore the other input. We consider three implementations of the module as follows:

**Best-Aligned Meaning (BAM).** This assessor only relies on the meaning $\mathcal{M}$ and uses the score of the most meaningful word discovered, $f_{bam}(\mathcal{C}, \mathcal{M}) = \max_{(w_j, \gamma_j) \in \mathcal{M}} \gamma_i$. This value represents the maximum possible alignment of this concept and a specific word. By our hypothesis, concepts that are highly aligned with a word should be meaningful and the corresponding word can be seen as a certificate for their meaning.

**Visual embeddings (VE).** This measure does not take the meaning into account but relies on the visual inputs only. Given the set of $n$ examples for a concept $\{x_i \mid i = 1 \ldots n\}$, we use embedding vectors $\{h_i \mid i = 1 \ldots n, \ h_i \in \mathbb{R}^{2048}\}$ that are extracted from a pretrained ResNet50 (He et al., 2016). To incorporate the concept activation $s_i$ we use an $s_i$-weighted version instead of plain Global Average Pooling (Lin et al., 2014) in the ResNet to compute the embedding $h_i$. We study two variants $f_{ve,mean}(\mathcal{C}, \mathcal{M}) = d_{mean}(\mathcal{C})$ and $f_{ve,min}(\mathcal{C}, \mathcal{M}) = d_{min}(\mathcal{C})$ that use coherence measures on the set of embeddings: $d_{mean}$ denotes the mean cosine between pairs of visual embeddings $h_i$ for a concept and $d_{min}$ the minimum cosine between two embeddings from the same concept.

**Hybrid assessors (HY).** It is also possible to combine different assessors to hybrid ones. We therefore investigate a simple weighted sum of multiple coherence assessors $f_i$, $f_{hy}(\mathcal{C}, \mathcal{M}) = \sum_i \lambda_i f_i(\mathcal{C}, \mathcal{M})$. We use a train-test split to fit the weights $\lambda_i$ on a data set of human assessments.

To compute the final score $q$ of a concept, we call both models sequentially, $q = f(\mathcal{C}, \text{Sem}(\mathcal{C}))$.

## 4 EVALUATION RESULTS

### 4.1 EVALUATION OF THE JOINT-VISION-LANGUAGE ANNOTATOR (JVL)

While the Object-Centric Annotator (OC) used in our work is an established model that we use in an off-the-shelf fashion without modification, we have to validate the functionality of the Joint Vision-Language Annotator (JVL). We start by comparing its provided annotations to labelled concepts in the Broden data set (Bau et al., 2017) containing 1197 frequent visual concepts. However, only some of the concepts come with activation maps (others are merely indicated as present or not), and we leave out concepts that have less than 3 examples with more than 8 % of the image salient for this concept. By this procedure, we arrive at a set of 572 concepts along with examples and corresponding activation maps. Additionally, the concepts are named by words, which we use as our ground truth and vocabulary ($|\Omega| = 572$). We run different annotation modules to see to what extent they can discover the true names. First, we compare our JVL module to a supervised baseline, that predicts the presence or absence of each concept and determines the final outcome by voting. Our JVL approach attains 72.6 % top-5 accuracy, compared to 61.7 % for the supervised model. Furthermore, we implement the annotation module devised by Laina et al. (2020) based on an image captioning model. Because its vocab is not restricted to Broden Concepts, we compare it to the JVL with 10000 frequent words as vocabulary. Our JVL obtains 37, 7 % top-5 accuracy versus 27, 2 % for the model by Laina et al. (2020) (cf. Appendix C.1 for details). We have thus verified that our Annotation Module is capable of identifying the most common visual concepts. It also outperformed other methods, which is why we only consider the JVL annotator and the OC annotator further.

### 4.2 CORRESPONDENCE TO HUMAN ASSESSMENT

**Evaluation Data Set Collection.** To evaluate the devised measures, we are looking for a diverse set of concepts of varying quality that will be assessed both by humans and our proposed framework for their meanings and coherence. We use two publicly available image data sets. Places365 (Zhou et al., 2017) consists of 365 categories of places encountered in everyday life and features 1.8 million training images (we use Places365-Standard). Animals with Attributes (AwA) 2 (Xian et al., 2018) was originally devised for zero-shot learning and contains 50 classes of animals and 37322 images. We do not make use of any attribute information in the data.

We train a supervised ResNet50 model on each data set and extract concepts using the ConceptSHAP approach (Yeh et al., 2019). Additionally, we use a simpler method that relies on k-Means clustering of image regions similar to Ghorbani et al. (2019) and Akula et al. (2020). For both approaches, we find the most salient patches by iterating through the data set and computing the activations. Because most of the concepts are meaningful to some extent, we artificially extend our data set with more

| | Coherence Assessor | | | |
| | Visual Embeddings (VE) | | Best-Aligned Meaning (BAM) | Hybrid (HY) combine (BAM ) + (VE) |
| Annotation Module | with $d_{min}$ | with $d_{mean}$ | - | $d_{mean}$ used for (VE) |
| Object Centric (OC) | $0.72\pm 0.02$ | $0.83 \pm 0.01$ | $0.70 \pm 0.02$ | $\mathbf{0.86 \pm 0.03}$ |
| Joint Vision-Language (JVL) | | | $0.39 \pm 0.02$ | $0.83 \pm 0.02$ |

Table 1: **Correlation score (Pearson's $\rho$) of our assessment with human rating for different annotators and assessors.** We obtain the highest score if we combine the BAM assessor using the OC annotation module with the best performing variant of the VE assessor.

negatives. These are created by duplicating an existing concept and randomly mixing in 25 % or 50 % of examples from other concepts on the same data set. We obtain 40 concepts with the method of Yeh et al. (2019) (the original hyperparameter), 50 from the k-Means clustering and 20 for each mix-in percentage. Doing so for each of the two data sets, this results in a total of 260 concepts.

**Collection of Human Assessments.** We collected 10 independent ground-truth meanings and assessments for each of the 260 concepts from 10 students at an average age of 24.9 ($\pm 2.1$). They were asked to assess the concepts given by $n = 18$ examples along with their activation maps. We chose this number because it fitted well on standard FullHD screens in a 3-by-6-array. First, raters were asked to give the best-fitting word for each concept. We provided the top-4 words from each of the semantic annotation modules as a choice, but also gave the raters the opportunity to give their own word in a text box. Subsequently, we asked the user to rate the fit of the concept to their (best possible) one-word description on a scale of four options between *not at all* (0) to *very well* (3).

**Evaluation of Coherence Assessors.** We implement the presented coherence assessment modules and evaluate their correspondence to human judgment. To this end, we compute the Pearson coefficient of correlation $\rho$ between the automated rating and the average one given by the users. Because some of the assessors (the hybrid models) have trainable parameters and our data set size is limited, we use 5-fold cross-validation and report the average results and the corresponding standard error. Results for 10 folds can be found in the supplement and are qualitatively equivalent.

Our results in Tab. 1 show that all three assessors are significantly correlated with the human rating. For the Best-Aligned Meaning (BAM), correlation using the OC annotator is higher ($\rho = 0.70$) than that obtained with the JVL annotator. We attribute this gap to the GradCAM activation maps for the words in the joint embedding, that are often not as precise as the detections by the OC annotator. The reduced supervision and increased generality apparently come at a price. We also observed that most of the descriptions the users chose for any concepts were indeed physical objects and seldomly more abstract concepts, thus testifying to the known importance of objects for human reasoning (Spelke & Kinzler, 2007). The assessments computed on mere visual embeddings (VE) show an even higher correlation with human judgment. However, it is possible to combine the BAM and VE assessors to a hybrid model by computing fixed weights for each assessor which leads to an increased correlation of $\rho = 0.86$ on the test set. This highlights that taking the semantic meaning into account can be beneficial for the assessment. Furthermore, the annotated words make the explanation easier to process compared to a set of examples.

## 5 DISCUSSION AND OUTLOOK

We discovered that object detections and visual embeddings allow for automatic assessment of concept coherence that exhibits significant correlation to human assessment. While the OC annotator provided objects and locations of higher quality, the JVL's larger scope makes it a strong competitor if more general annotations are required. We hypothesize that with better vision-language embedding models and localization maps, the correlation with the human rating can further be improved. Overall, we believe our work can be impactful in many domains, as it is among the first to quantify the degree of interpretability automatically and on the semantic level. We therefore think our SOLaCE framework will help lead the way towards more user-friendly AI systems.

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

## A    ADDITIONAL RELATED WORK

**Semantic coherence in topic modeling.**    Semantic coherence is an established research area in topic modeling for natural language texts and dialogues (Mimno et al., 2011; Röder et al., 2015; Vakulenko et al., 2018). Many works propose different coherence measures of which a variety are listed for example in the works of Röder et al. (2015) and Zupanc & Bosnic (2014). In our paper, we attempt to transfer the insights from these works to the vision domain.

**Joint Vision-Language Models.**    With the recent advancement in language (Brown et al., 2020) and vision-language models (Radford et al., 2021; Changpinyo et al., 2021) trained on extremely large and generic data sets, in this paper, we investigate whether such a model can provide the necessary supervision to quantify the semantic coherence of concepts. Our work exhibits some similarities with image captioning (Karpathy & Fei-Fei, 2015; Anderson et al., 2018; Hossain et al., 2019) and group captioning in particular (Li et al., 2020). We build on this work to obtain an interpretability score that corresponds to human assessment, which has not been investigated in this context.

**Meaningfulness Evaluation of Visual Concepts.**    Kraft et al. (2021) propose a variety of evaluation measures for automatically discovered prototypes, relying on their alignment with ground truth objects or parts. Using a set of single words as a meaning has been proposed as an evaluation by Ghorbani et al. (2019), but the words are still given by human annotators. On the contrary, Hernandez et al. (2022) recently proposed a powerful annotation module to assign natural language descriptions to deep features (which can be viewed as concepts), but did not run a coherence evaluation based on these descriptions. In this work, we aim to bridge this gap and combine both steps sequentially.

## B    IMPLEMENTATION

### B.1    APPLICABILITY OF OUR DEFINITION

For manually or algorithmically annotated concepts in the form of segmentation masks, such as in the Broden data set (Bau et al., 2017), the activation value for the pixel is set to 1 if the corresponding pixel is annotated for a certain concept and to 0 otherwise. Furthermore, activations of feature maps in convolutional neural networks (CNNs) can be upsampled to the image resolution and normalized to match our framework. Thereby, concepts learned by convolutional filters can be considered. Works that define concepts as vectors in the latent space (Ghorbani et al., 2019; Yeh et al., 2019), can be put to the test by computing the similarity (commonly cos-similarity) of the latent vector for each patch with the concept vector (cf. Yeh et al. (2019) for details).

### B.2    OBJECT-CENTRIC ANNOTATION MODULE (OC)

For the supervised annotation, we draw inspiration from the object detection model shown by Anderson et al. (2018), that can detect both objects and attributes. We preprocess the VG (Krishna et al.,

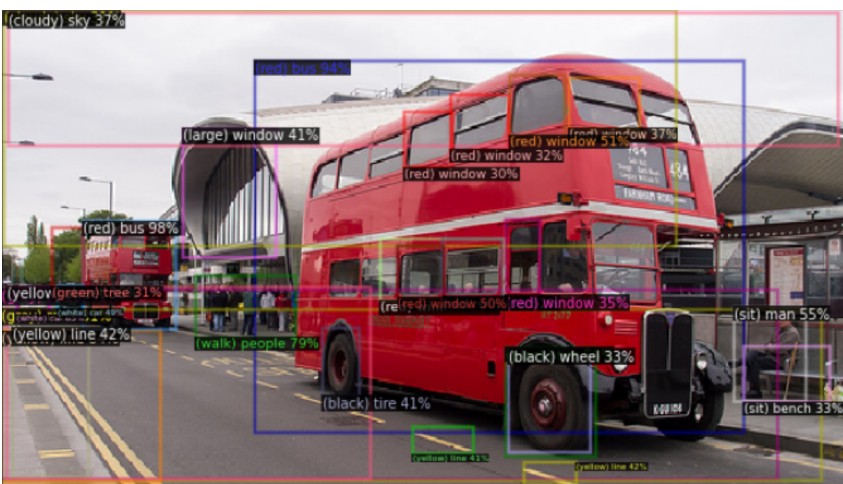

Figure 3: Detections from FasterRCNN on a randomly chosen example for the from the VG test set. The predicted attribute is shown in brackets before the object class but not used in the final annotator.

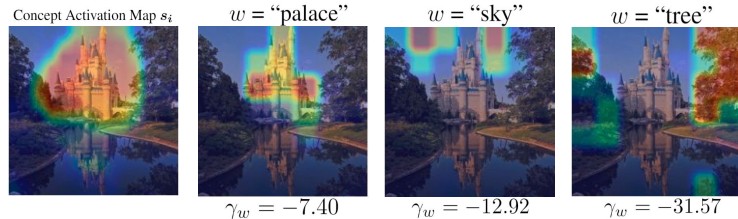

Figure 4: **Visual-semantic alignment in the zero-shot annotation module.** The example image corresponds well to multiple words. Thus, to find out the correct one, the concept activation map $s_i$ (first picture) has to align with the activation $\boldsymbol{\alpha}_{i,w}$ of candidate words $w$ (each of the following pictures). We see that higher alignment leads to better scores $\gamma_w$ (cf. Eqn. 2).

2017) data set and aggregate the object categories and attributes according to the WordNet synsets provided with the data set. Because classes with few observations are almost never predicted, we remove attributes and objects with less than 8 occurances. By doing so, we arrive at 3434 (before: 7842) object classes, and 2979 (before: 6277) attributes. We then train the Faster-RCNN provided by Detectron2 (Wu et al., 2019), with the additional attribute branch proposed by Anderson et al. (2018). We use Detectron2's default settings, with only two changes: We employ a small learning rate of $1 \times 10^{-4}$ that is further decreased by a factor of 10 after 80k and 90k iterations. We observed training to be sufficiently converged after 100k iterations. Because wrongly detected objects seemed to harm the assessment less that objects that were not detected at all, we used a low detection threshold of $0.3$ which results in many detections. We experimented with the attribute information as well, but did not use it in the final experiments in this paper. The output of the predictor can be seen in Fig. 3.

We use the class names as our dictionary $\Omega$ and use the Intersection over Union (IoU) score (inspired by Bau et al. (2017)) between the object bounding boxes and the concept activation as alignment score. The final score $\gamma_w$ for a word $w$ is given by the sum of the IoUs over all concept examples. Therefore, let $\boldsymbol{b}_{i,w} \in \{0,1\}^{H \times W}$ be a binary vector indicating the pixels that are in the bounding boxes for a detected object of type $w$ in image example $i$. The final score $\gamma_w$ for a word $w$ is given by the sum of the IoUs over all concept examples:

$$\gamma_w = \frac{1}{|\mathcal{C}|} \sum_{(\boldsymbol{x}_i, \boldsymbol{s}_i) \in \mathcal{C}} \frac{\sum \min(\boldsymbol{s}_i, \boldsymbol{b}_{w,i})}{\sum \max(\boldsymbol{s}_i, \boldsymbol{b}_{w,i})} \tag{1}$$

The $\min$ and $\max$ operations are applied pixel-wise and summation is done over all pixels.

### B.3 Joint Vision-Language Annotation Module (JVL)

For JVL model, we use the pretrained CLIP model (Radford et al., 2021) with the ResNet50 backbone, because it provided the most reliable word activation maps. To come up with an extensive set of possible candidate works, we fuse several sources to obtain a powerful dictionary $\Omega$:

1. Google 10000 frequent words [2]
2. VG list of classes [3]
3. Most common nouns in English [4]

We only keep nouns in the source 1 (by applying the tagger from WordNet). During development of user surveys, we observed that verbs or adjectives were almost never chosen. Additionally, we think that it is fairer to compare to the supervised FasterRCNN that also uses only nouns. In total, $|\Omega| = 10037$ for the experiments. The exact words are provided with our data set. We compute the scores $y_{i,w}$ for all words and the images pairs, which can be done very efficiently, because of the form of the zero shot model: $y_{i,w} = J(\boldsymbol{x_i}, w) = J(\boldsymbol{u}(\boldsymbol{x_i}), \boldsymbol{v}(w))$, where $\boldsymbol{u} \in \mathbb{R}^d$ is a visual embedding and $\boldsymbol{v} \in \mathbb{R}^d$ a text embedding. For the CLIP model we use, $d = 512$. The vectors have the same dimensionality, and simple dot-products can provide the similarity score $(\boldsymbol{u} \cdot \boldsymbol{v})$, that is normalized via a softmax function. Thus, for all words and images, we have to compute their latent representations only once, and can compute the scores by a matrix multiplication $\boldsymbol{Y} = \boldsymbol{U}\boldsymbol{V}^\top$, where $\boldsymbol{Y} \in \mathbb{R}^{n \times |\Omega|}$ holds all the scores, $\boldsymbol{U} \in \mathbb{R}^{n \times d}$ contains one visual embedding per line, and $\boldsymbol{V} \in \mathbb{R}^{|\Omega| \times d}$ holds one word embedding per line. Because the dictionary is the same for all concepts, $\boldsymbol{V}$ can be precomputed beforehand. Thus, to compute all scores for a concept, it is only necessary to compute the visual embeddings $\boldsymbol{U}$ of the examples, and matrix-multiply them with the present word-vectors.

For the more expensive alignment step however, we do not keep all words after this stage, but preselect a set of promising candidates based on the $y_{i,w}$. For each image $\boldsymbol{x_i}$ in the concept, we add the top $p$ words with the highest similarity scores $y_{i,w}$ to a set of candidate words. This results in at most $p \cdot n$ candidates though fewer are returned in practice due to words reappearing. The other words are not further considered.

To align visual and semantic activations, we also estimate an activation map $\boldsymbol{\alpha}_{i,w} \in \mathbb{R}^{H \times W}$ (normalized to $[0, 1]$) for each candidate word using GradCAM (Selvaraju et al., 2017) on the model $J$. The final score of a word is given by the negative Mean-Squared-Error (MSE) between all concept and word activation maps

$$\gamma_w = -\frac{1}{|C|} \sum_{(\boldsymbol{x_i}, \boldsymbol{s_i}) \in \mathcal{C}} \mathrm{MSE}\left(\boldsymbol{s_i}, \boldsymbol{\alpha}_{i,w}\right). \tag{2}$$

The example in Figure 4 illustrates this approach. CLIP can work with entire sentences as inputs as well, thus we tried to use prompts such as "There is a ... in the image" or "This is an image of a ..." incorporating with different words $w$ from the dictionary. Nevertheless, we did not observe much performance improvement (better alignment scores) than using singles words. Therefore, we decided to use words only for the final JVL implementation.

### B.4 Visual Embeddings Coherence Assessor (VE)

The visual embeddings we use in this work stem from a ResNet50 model pretrained on ImageNet, that comes with the PyTorch framework. In this specific architecture, after the forth (and final) residual layer, the input is transformed into a tensor $\boldsymbol{\Phi}$ of dimensionality $\boldsymbol{\Phi} \in \mathbb{R}^{r \times r \times d}$ (given an input size $448 \times 448$, $r = 14, d = 2048$). Usually, this tensor is global average pooled (this is mathematically equivalent to taking the mean values) over the first two dimensions, resulting in a feature vector $\boldsymbol{h}_i \in \mathbb{R}^{2048}$. To bring in the locality information $\boldsymbol{s}_i$, we do not take the mean of the

---

[2] We used the document named "google-10000-english.txt" under the link https://github.com/first20hours/google-10000-english
[3] https://visualgenome.org/data_analysis/statistics
[4] https://7esl.com/list-of-nouns/

spatial dimensions, but resample the activation map to the same resolution as the feature vectors before the pooling (e.g., $\hat{\boldsymbol{s}}_{\boldsymbol{i}} \in \mathbb{R}^{r \times r}$). We further normalize $\hat{\boldsymbol{s}}_{\boldsymbol{i}}$ to sum up to 1 and take the the final feature vector to be

$$\boldsymbol{h}[t] = \sum_{l=1}^{r} \sum_{m=1}^{r} \hat{\boldsymbol{s}}_{\boldsymbol{i}}[l, m] \boldsymbol{\Phi}[l, m, t], \forall t = 1 \ldots d, \tag{3}$$

a spatially-weighted average. For the visual coherence assessor we use the following measures $f_{v,mean}$,

$$f_{v,mean}(\mathcal{C}, \cdot) = \frac{1}{n^2} \sum_{i=1}^{k} \sum_{j=1}^{k} \frac{\boldsymbol{h}_i \cdot \boldsymbol{h}_j}{\|\boldsymbol{h}_i\| \|\boldsymbol{h}_j\|}, \tag{4}$$

which returns the average cosine between cluster examples and the minimal cos of two cluster members,

$$f_{v,min}(\mathcal{C}, \cdot) = \min_{i,j=1 \ldots n} \frac{\boldsymbol{h}_i \cdot \boldsymbol{h}_j}{\|\boldsymbol{h}_i\| \|\boldsymbol{h}_j\|}. \tag{5}$$

## C  ADDITIONAL EVALUATION RESULTS AND DETAILS

### C.1  EVALUATION THE JOINT VISION LANGUAGE ANNOTATION MODULE (JVL)

In this section we provide details on how we confirmed the functionality of our Joint Vision Language (JVL) Annotation Module on the Broden data set.

First, we had to conduct a preprocessing step. Because the Broden data set is a fusion of several sources, some labels appear in different forms of synonyms (e.g., {bus, autobus}), singular and plural (e.g., {door, doors}) and hyponyms and hyperomyms (e.g., {table, coffee table}). We manually merge those words together into 449 sets before running the experiment and consider the prediction as correct, if the word proposed is in the same merged set as the ground truth. We provide the list of sets along with our code, so that it remains possible to reproduce our findings and compare new results to ours. We consider the following models to assign $k = 5$ words (with descending scores) to sets of examples from Broden:

**JVL (Ours).**  We implement our JVL annotation model with the proposed saliency map alignment with CLIP (ResNet50 backbone) as embedding model. We use a general purpose vocabulary ("Full Dict", $|\Omega|=10037$) with the most frequent English nouns and a special purpose vocabulary that contains the names of the Broden concepts ("Domain Dict", $|\Omega|=572$).

**CLIP.**  We compare our results against a plain implementation of CLIP without our alignment step. This model just returns the word with the highest summed score $\gamma_w = \sum_i y_{i,w}$. We use the same two dictionaries.

**Supervised Baseline.**  Furthermore, we implement a supervised baseline model for multilabel classification on the Broden data that predicts presence or absence of each of the 572 concepts for each example image. The final concept-level predictions are found by summing up the predictions (ground truth concept absent or present) for each example image and returning the concept names with the highest summed prediction score.

**Laina et al.**  Finally, we put the group captioning model devised by Laina et al. (2020) to the test. It is trained on the Conceptual Captions data set (Sharma et al., 2018) as in the original paper and outputs entire sentence captions for a group of images. We count the prediction as correct, if the ground-truth name of the concept appears in the sentence predicted. Note that this is a slight advantage over other models as there are several words in the caption.

For each ground truth concept, we sample sets of $n = 30$ examples (or use all images if there are not enough) and compare the prediction of the model to the ground truth names. Results of the experiment on Broden can be found in Table 2. We first observe that our saliency alignment ("JVL") can considerably improve the top-1 and top-5 score over the plain CLIP for both vocabularies. This highlights the benefit of our suggested method to incorporate locality information. The general

| Annotation Module | Top-5 Acc. | Top-1 Acc. |
|---|---|---|
| CLIP (Domain Dict) | $58.3 \pm 0.2\,\%$ | $36.7 \pm 0.1\,\%$ |
| Supervised Baseline | $61.7 \pm 0.5\,\%$ | $27.4 \pm 0.2\,\%$ |
| JVL (Domain Dict) | $\mathbf{71.2} \pm 0.2\,\%$ | $\mathbf{48.6} \pm 0.2\,\%$ |
| CLIP (Full Dict) | $20.6 \pm 0.2\,\%$ | $7.8 \pm 0.1\,\%$ |
| Laina et al. | $27.2 \pm 0.2\,\%$ | $8.5 \pm 0.3\,\%$ |
| JVL (Full Dict) | $\mathbf{37.7} \pm 0.5\,\%$ | $\mathbf{19.6} \pm 0.2\,\%$ |

Table 2: **Agreement of the JVL annotation module with ground-truth annotations for the Broden data set.** Top-5 and top-1 accuracy are reported for the plain CLIP module (no alignment with concept activation map), the model by Laina et al. (2020), a supervised predictor, and our JVL model. Our JVL Model performs best at correctly identifying the true labels in with the more general setting, and when restricted to the domain vocabulary.

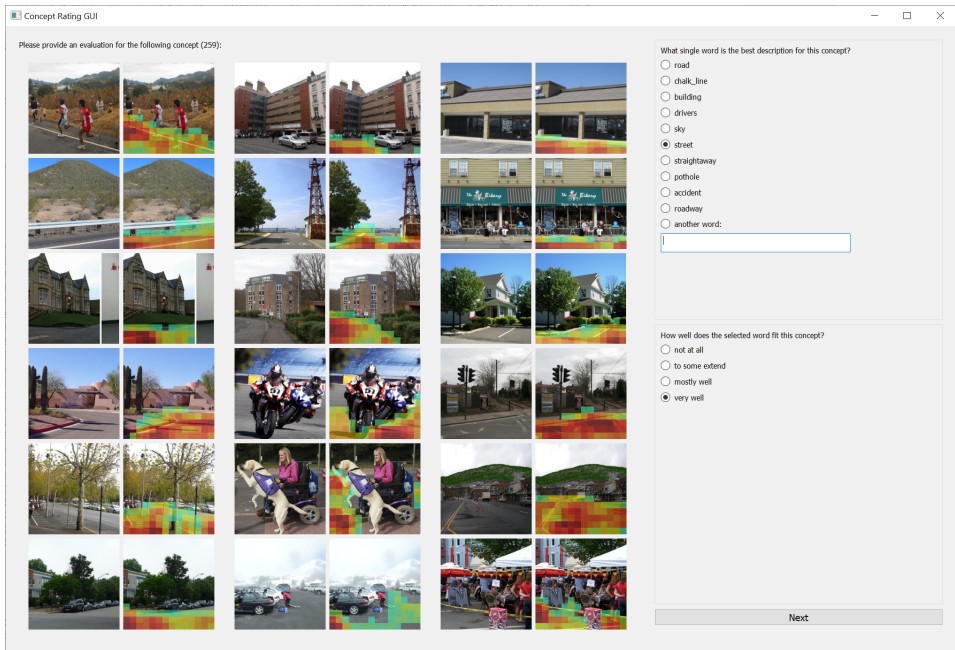

Figure 5: The user interface for our study.

purpose captioning model by Laina et al. is listed between the plain CLIP and our method. Perhaps unsuprisingly, the accuracy can be much improved by using the domain specific-vocabulary. In this setting, our method can even outperform the supervised model. Overall, we see that the actual meaning is listed among the top five words in about 70 % of the cases and comes in first in about half of the cases. Considering the fine-grained nature of the problem (500+ classes and no finetuning), these are highly respectable results that prove our annotation module to be powerful enough to detect common visual concepts.

## C.2 Agreement with human Assessment

In this last section, we conduct an extended analysis of our human assessment data set. The user interface for our study is shown in Fig. 5.

**Probing the data set.** First, we report the average ratings by method and data set in Tab. 3. We observe that the mix-in concepts get a lower ratings than the actual concept discovery algorithms which further decreases when more random images are mixed in. The data set thus passes this basic sanity check. The concepts on the Places365 data set are better received than those on AwA. They

| | Method/Dataset | Places365 | AwA | $N$ |
|---|---|---|---|---|
| **Average Ratings** | Yeh et al. | **2.22** | 1.34 | 400 |
| | k-Means | 2.20 | **1.44** | 500 |
| | 25 % mix-in | 1.45 | 1.10 | 200 |
| | 50 % mix-in | 0.90 | 0.90 | 200 |

Table 3: **Mean human ratings (0: worst, 3: best)** for concepts from different methods ("Yeh et al.", "k-Means") on two datasets ("Places365", "AwA"). $N$ denotes the total number of human ratings per dataset and "mix-in" gives a percentage of artificially randomized samples per concept. Ratings on unaltered concepts are significantly better than with randomized samples.

| | Coherence Assessor | | | |
|---|---|---|---|---|
| | Visual Embeddings (VE) | | Best-Aligned Meaning (BAM) | Hybrid (HY) combine (BAM ) + (VE) |
| Annotation Module | with $d_{min}$ | with $d_{mean}$ | - | $d_{mean}$ used for (VE) |
| Object Centric (OC) | $0.73\pm 0.02$ | $0.83 \pm 0.01$ | $0.71 \pm 0.02$ | $\mathbf{0.86 \pm 0.03}$ |
| Joint Vision-Language (JVL) | | | $0.37 \pm 0.02$ | $0.83 \pm 0.02$ |

Table 4: **Correlation score (Pearson's $\rho$) of our assessment with human rating for different annotators and assessors.** This table corresponds to Tab 1 but using 10-fold cross-validation.

often focus on specific objects or scenes. We see that the method devised by Yeh et al. (2019) fares marginally better on the Places365, but is outperformed by the simpler k-Means approach on AwA.

**Agreement between raters.** We observed that rating the coherence of a concept can be a highly subjective task. Therefore, we investigate to what extent the raters agree on an evaluation. We run a test on the variance of our ratings. In our null hypotheses, all ratings are indenpendently drawn from the overall distribution of ratings, with sample standard deviation $\sigma_s = 0.93$. We then compute the sample standard deviation of the ratings for a single concept and perform a Chi-Squared test, to check if it is significantly smaller than $\sigma_s$. If the variance of the ratings is significantly smaller, than it would be in a random setting, this indicates an above-random agreement among the raters. Using $p = 0.01, p = 0.05$, we obtain a a-random above-chance agreement for 58 % and 82 % of the concepts respectively. We also compute Cohan's $\kappa$ for multiple item agreement. Note however that this measure neglects the order of the items and treats all disagreements (e.g., the ratings (0,3) and (0,1)) equally. We get a value of $\kappa = 0.25$ which testifies an agreement above chance level, but indicates also frequent disagreement among raters.

**Results under 10-fold cross-validation.** In Table 4 we provide the correlations between by our assessment modules and the human ratings using 10-fold cross-validation. The qualitative results are the same as in our main paper.

**Scatter Plots.** Because linear correlations can be overconfident, we show scatter plots of the results obtained with our assessors (BAM, VE) and the corresponding ground truth rating in Fig. 6 (left). We observe that the correlation can be well modelled as linear and that it is of significant strength. Finally, we wish to highlight that is is also possible to filter out non-meaningful concepts well, which is testified by the corresponding ROC-curves (right).

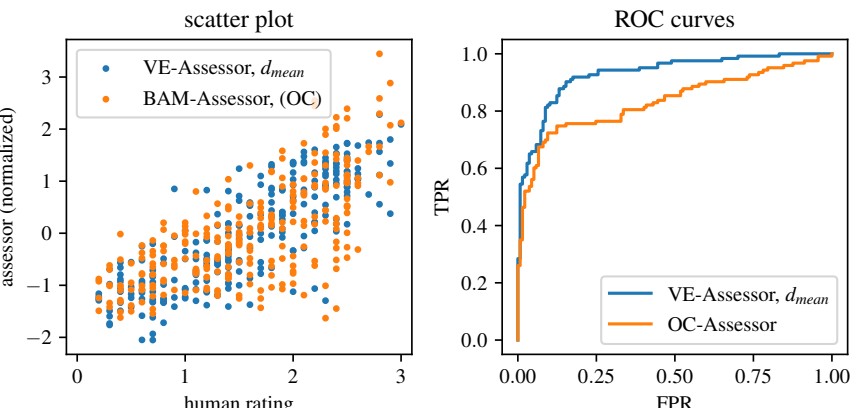

Figure 6: Scatter plots of our automatic assessments and the corresponding human ground truth (left). We show the ROC curves for deciding whether a concept is of higher or lower intelligibility than the median (right). This figure highlights that Coherence Assessment Modules can be used to separate the meaningful from non-meaningful visual concepts.

