# OpenReview forum: "Coherence Evaluation of Visual Concepts With Objects and Language"
_ICLR.cc/2022/Workshop/OSC — ICLR2022 OSC  Poster_

### Official Review · Reviewer_BLFE · 2022-03-15
**Good workshop paper. Looking at an interesting direction.**

**Rating:** 2
**Confidence:** 3

**Review:**

This paper works on building an automated evaluation toolkit for unsupervised/weekly supervised visual concept discovery. Specifically, the  proposed method (SOLaCE) evaluates visual concepts discovered by image classification models trained on higher-level classification tasks. For example, the model is trained on classifying scenes and the model automatically discovers different object types.

The proposed approach is well-motivated, as stated in the introduction section, "meaningful visual concepts have concise descriptions in natural language." Thus, the paper proposes to leverage pre-trained visual-language models (CLIP) to score proposed concepts. The paper has also presented a human study on how their automatic evaluation agrees with human judgement. Overall the model is well-presented, relevant to the theme of the workshop, and can be a good contribution to the workshop.

Below I listed a few weaknesses of the paper.

1. The authors have primarily motivated the work by discussing generally "language descriptions," however, as shown in Table 1 (also stated by the author in their results analysis paragraphs), the JVL model (CLIP-based) actually performs worse than the simple object-detection based model.
2. The concepts used for training the Visual Genome model may not align well with the AVA dataset the authors used. Also, most of the visualizations of the results are presented on scene365. The authors are encouraged to analyze more results on the AVA dataset.
3. The authors have stated that the "CLIP model is more general," however there is no experimental support for that (e.g., application to domains with more concepts).
4. The paper has referred to cognitive science arguments (human XXXXXX) in various places. It could be better if the authors can cite relevant papers and draw explicit connections between their arguments and results documented in cog-sci studies.

---

### Official Review · Reviewer_VzPh · 2022-03-16

**Rating:** 2
**Confidence:** 3

**Review:**

This paper proposes a method to automatically score the quality of “visual concepts” extracted by unsupervised methods (usually in the form of a set of images and associated activation maps).

This is done by combining a network which produces labels for activated image regions, as well as a scoring function. The paper assesses a few ways to produce labels (Faster R-CNN to generate object bounding boxes; CLIP to directly compute language-visual consistency scores), as well as a few scoring functions, and they show that these correlate well with human judgments.

Although this targets a very specific problem, I found the paper easy to follow and making clear decisions throughout, so it might be of interest to the workshop’s attendees.

Questions and comments:
1. Did you directly use the single word labels for CLIP as it appears in Figure 2? Did you explore more advanced prompt hacking? I would have expected this to have quite a strong effect, considering what the community has been exploring recently (see also DALL-E re-ranking and co)
2. Wouldn’t the BAM require the gammas to be calibrated/normalized over the whole “data” in order for the max to be well-behaved?
3. Instead of Table 1 with Pearson’s correlation scores, I would have preferred to see the raw scatter plots and judge for myself how correlated the scores and human ratings would have been (I do not trust the linearity assumption and these are too easily overconfident).
   1. Could you add these to the Appendix perhaps?
4. The fact that the CLIP results depend on yet another way to extract activation maps is slightly unfortunate, especially as the authors mention that the Grad-CAM activation quality might be causing issues.
   1. Are there particular avenues to fix that issues or alternative scores which would bypass this issue?

---

### Decision · Program_Chairs · 2022-03-23

Accept (Poster)